# Increased neutralization and IgG epitope identification after MVA-MERS-S booster vaccination against Middle East respiratory syndrome

Anahita Fathi [1,2,3,4], Christine Dahlke[1,2,3], Verena Krähling[5,6], Alexandra Kupke[5,6], Nisreen M. A. Okba [7], Matthijs P. Raadsen[7], Jasmin Heidepriem [8], Marcel A. Müller [9,10], Grigori Paris[8], Susan Lassen[1,2,3], Michael Klüver [5,6], Asisa Volz[11,12], Till Koch [2,3,4], My L. Ly[1,2,3], Monika Friedrich[1,2,3], Robert Fux[13,14], Alina Tscherne[13,14], Georgia Kalodimou[13,14], Stefan Schmiedel[3,4], Victor M. Corman [9,10], Thomas Hesterkamp[15], Christian Drosten [9,10], Felix F. Loeffler [8], Bart L. Haagmans [7], Gerd Sutter[13,14], Stephan Becker[5,6] & Marylyn M. Addo [1,2,3✉]

Vaccine development is essential for pandemic preparedness. We previously conducted a Phase 1 clinical trial of the vector vaccine candidate MVA-MERS-S against the Middle East respiratory syndrome coronavirus (MERS-CoV), expressing its full spike glycoprotein (MERS-CoV-S), as a homologous two-dose regimen (Days 0 and 28). Here, we evaluate the safety (primary objective) and immunogenicity (secondary and exploratory objectives: magnitude and characterization of vaccine-induced humoral responses) of a third vaccination with MVA-MERS-S in a subgroup of trial participants one year after primary immunization. MVA-MERS-S booster vaccination is safe and well-tolerated. Both binding and neutralizing anti-MERS-CoV antibody titers increase substantially in all participants and exceed maximum titers observed after primary immunization more than 10-fold. We identify four immunogenic IgG epitopes, located in the receptor-binding domain (RBD, $n = 1$) and the S2 subunit ($n = 3$) of MERS-CoV-S. The level of baseline anti-human coronavirus antibody titers does not impact the generation of anti-MERS-CoV antibody responses. Our data support the rationale of a booster vaccination with MVA-MERS-S and encourage further investigation in larger trials. Trial registration: Clinicaltrials.gov NCT03615911.

A full list of author affiliations appears at the end of the paper.

Emerging infections pose a major threat to public health. In recent years, outbreaks of severe acute respiratory syndrome (SARS), Ebola virus disease (EVD) and, currently, coronavirus disease 2019 (COVID-19) have been declared public health emergencies of international concern. In response, international organizations such as the World Health Organization (WHO) and the Coalition of Epidemic Preparedness Innovations (CEPI) have developed guidance for research and development to increase pandemic preparedness, and swift and effective vaccine development plays a central role. It includes two strategic elements: The advancement of vaccine platforms, which can serve as blueprints for vaccine candidates against newly emerging pathogens, as well as the development of vaccine candidates against diseases likely to cause future epidemics, defined as priority diseases by the WHO[1].

Viral vectors represent promising vaccine platforms. They comprise recombinant attenuated or replication-deficient viruses that express gene sequences of the pathogen of interest. Vector vaccines are being evaluated as vaccine candidates against various emerging pathogens[2–4], and have received licensure as vaccines against EVD[5,6] and COVID-19[7,8]. Modified Vaccinia virus Ankara (MVA) is a well-established replication-deficient poxviral vector that does not integrate into host cell DNA[9]. Non-recombinant MVA has been licensed as a smallpox vaccine, and recombinant MVA has been studied as a vector vaccine candidate for multiple infectious disease and cancer indications[10]. It has been administered to over 120,000 individuals, including immunocompromised populations and children[6,11], and has demonstrated a favorable safety profile[9,12]. In addition, a recombinant MVA-based vaccine has been licensed as part of a heterologous vaccination regimen against EVD[5].

MVA-MERS-S is a novel MVA-based vaccine candidate against the Middle East respiratory syndrome coronavirus (MERS-CoV) encoding its full spike glycoprotein (S). The coronaviruses' spike glycoprotein has been readily identified as a highly immunogenic viral structure that is the main target of most coronavirus vaccine candidates (i.e., vaccines against MERS-CoV, SARS-CoV, and SARS-CoV-2)[13,14]. Necessary for viral entry, it consists of the subunits S1 and S2, where S1 contains the functionally relevant receptor-binding domain (RBD), which mediates host cell binding and attachment and S2 mediates membrane fusion via its fusion glycoproteins and the transmembrane domain.

MERS-CoV, alongside SARS-CoV and SARS-CoV-2, belongs to the group of highly pathogenic beta-coronaviruses that have been identified as priority pathogens[1]. Infection may lead to pneumonia and multi-organ failure, and is associated with a case-fatality rate of up to 35%[15]. Since its emergence in 2012, MERS-CoV has caused multiple outbreaks and has been exported to 27 countries. Apart from MVA-MERS-S, one DNA[16] and one adenoviral vector vaccine candidate[17] against MERS-CoV have completed Phase 1 trials. However, there is still no licensed vaccine or specific therapy available.

In 2017–2018, we conducted a single-center, open-label Phase 1 clinical trial of MVA-MERS-S in 23 healthy men and women aged 18–55 years (EudraCT No. 2014-003195-23, ClinicalTrials.gov Identifier NCT03615911)[18]. Participants received two injections of either $1 \times 10^7$ plaque-forming units (PFU; low dose: LD) or $1 \times 10^8$ PFU (high dose: HD) MVA-MERS-S in a homologous regimen on Days 0 and 28. Safety, tolerability, and immunogenicity were evaluated until the end of the study on Day (D) 180. The regimen was well-tolerated; only transient reactogenicity was observed and no severe or serious adverse events (SAE) occurred. Both cellular and humoral immune responses were elicited after two vaccinations. Seroconversion occurred in 75% of LD and 100% of HD participants, peaked on D42, and

binding antibody (Ab) titers correlated with neutralizing antibody (nAb) levels. However, it is not known whether booster vaccinations with MVA-MERS-S are necessary to induce optimal immunogenicity.

In this study, we examine the safety profile and the immunogenicity of a booster vaccination in a subgroup of the original MVA-MERS-S trial participants one year ($12 \pm 4$ months) after primary immunization. We find that MVA-MERS-S booster vaccination is safe and substantially increases both MERS-CoV-specific binding and neutralizing antibody titers. MVA-MERS-S vaccination induces antibody responses to four S1 and S2 IgG epitopes, including one epitope previously described to be the target of an anti-MERS-CoV monoclonal antibody found in convalescent individuals that showed protective capacity in mice. The results presented here contribute to our understanding of the MVA-MERS-S vaccine platform and support the further assessment of MVA-MERS-S booster vaccinations, which are currently evaluated in a larger Phase 1b trial (NCT04119440).

## Results

**Study overview.** The initial trial assessing primary immunization of MVA-MERS-S (including recruitment and a 180-day observation period) was conducted between Dec 17, 2017, and June 5, 2018. We subsequently initialized the booster extension study, which was conducted between March 13th, 2019 and April 15th, 2019, as a single-center openlabel trial in Hamburg, Germany. A study overview is given in Fig. 1. All individuals who had received two primary immunizations with MVA-MERS-S ($n = 23$) were informed about the study extension via phone call and invited to participate in the booster extension trial. Ten trial participants agreed to participate, were re-screened, and were recruited to receive a third (booster) vaccination with MVA-MERS-S one year ($12 \pm 4$ months) after primary immunization. Three individuals had been primed with the LD and seven with the HD on Days 0 and 28. As the higher dose ($1 \times 10^8$ PFU) of MVA-MERS-S had proven to be safe and immunogenic, all participants received this dose independent of initial LD/HD group. The booster vaccinations were administered between March 13th and 18th, 2019 (Boost Day 0; B:D0) and the participants returned to the study site for immunogenicity and/or safety assessments on B:D1, B:D3, BD:7, and B:D14. All participants completed the study on B:D28. Supplementary Table 1 provides an overview of participant characteristics. In addition, healthy unvaccinated individuals served as a control group for immunogenicity analyses ($n = 2–6$, depending on the assay). Analyses on pre-existing immunity to other common-cold human coronaviruses (HCoV) were performed on all participants of the original Phase 1 MVA-MERS-S clinical trial.

**Safety profile of the booster vaccination.** Nine participants (90%) experienced at least one adverse event (AE) following the booster vaccination. In total, 51 AEs were recorded, of which 40 were classified as related and 11 as unrelated to the vaccine. The most common AE was local reactogenicity, experienced by 9/10 vaccinees, followed by fatigue/malaise (6/10 vaccinees). The majority of related AEs (80%, 32/40) were mild (grade 1) and solicited (92.5%, 37/40). Only one participant, who was in the former LD cohort, experienced grade 3 AEs one day after booster vaccination, all of which resolved spontaneously within 24 h. No serious adverse events (SAEs) were recorded. AEs occurred early after vaccination (median 1 day, interquartile range (IQR) 0.25–1) and were transient (median duration 1 day, IQR 0–2.75).

In order to compare the safety profile of primary and booster vaccinations, we analyzed the AE frequency, duration, and quality after primary vaccination in this subgroup. Related AE had

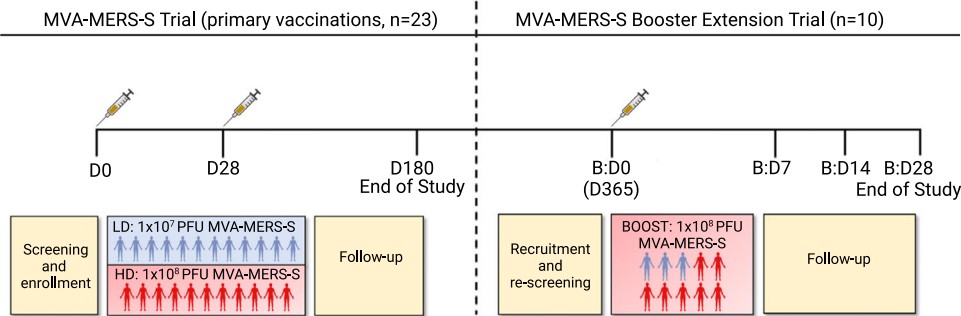

**Fig. 1 Study design and trial profile.** Twenty-three individuals completed a homologous primary immunization with two vaccinations (Day (D) 0 (D0) and D28) of either $1 \times 10^7$ plaque-forming units (PFU) (low dose (LD), blue) or $1 \times 10^8$ PFU MVA-MERS-S (high dose (HD), red), and were followed-up for 180 days, which concluded this part of the study (End of Study). Participants were invited to return for an additional third vaccination as a booster 1 year ±4 months after prime. 10 participants (3 from the LD, 7 from the HD group) were re-enrolled and received a dose of $1 \times 10^8$ PFU MVA-MERS-S. Safety and tolerability were assessed on Boost (B) Days (B:D) 0 (B:D0, baseline), B:D1, B:D3 (not depicted), B:D7, B:D14, and B:D28; humoral immunogenicity was assessed on B:D0, B:D7, B:D14, and B:D28 (End of Study). All 10 participants completed the extension trial and were included in the analyses. Created with BioRender.com.

occurred in 8/10 (80%) of individuals after primary vaccination. As seen after booster vaccination, the median time to AE occurrence after the first vaccination had been 1 day (IQR 1–1) with a median AE duration of 1 day (IQR 1–4). AE had likewise been predominantly solicited (46/49 AE, 94%) and mild (46/49 AE, 94%), suggesting a comparable safety profile of primary and booster vaccinations. A detailed overview of AE frequency and grade after the booster vaccination can be found in Supplementary Table 2. In addition, Supplementary Fig. 1 provides an overview of AE type and grade of the three vaccinations in the individuals who had also received the booster dose ($1 \times 10^8$ PFU) as the primary regimen (HD).

As a result of the intensive study schedule, including early study visits and comprehensive laboratory analyses, we were able to closely assess hematologic changes and markers of organ function after booster vaccination. We observed transient changes in leukocyte counts with an increase by B:D1 and a decrease by B:D3 (Fig. 2a), which could be attributed to a respective dynamic in neutrophil counts (Fig. 2b). Thrombocyte counts transiently decreased by B:D1 (Fig. 2c). The changes of leukocyte, neutrophil and thrombocyte counts largely remained within the range of physiologic variation. A decrease in circulating lymphocytes on B:D1 (Fig. 2d) as well as a discreet increase in plasma C-reactive protein (CRP) levels on B:D1 and B:D3 (Fig. 2e) were also observed. None of these changes were clinically significant, however, they indicate biologic activity of the vaccination and suggest cellular redistribution to other compartments. By B:D7, no hematologic changes or CRP elevation were observed as compared to baseline. All other measured biomarkers remained unchanged (i.e., markers of renal, hepatic, pancreatic, and cardiac organ function/injury, and electrolytes).

We compared the relative changes of CRP and hematologic parameters after booster vaccination to those observed after primary vaccinations (Supplementary Fig. 2). In the group of individuals who had received the booster dose ($1 \times 10^8$ PFU) also as primary vaccinations ($n = 7$, HD), we observed a less pronounced decrease of thrombocyte counts from baseline after second and booster vaccination as compared to first vaccination, while relative neutrophil and, hereby, whole leukocyte counts, were slightly increased after booster vaccination compared to primary vaccinations. This increase in neutrophils was highly correlated ($p < 0.0001$) with leukocyte counts and inversely correlated ($p = 0.0042$) with lymphocyte counts, further underlining the biologic effect of the booster vaccination. The hematologic changes were, however, less than two-fold (physiologic range up to three-fold, see also Fig. 2).

**Humoral immunogenicity of a late booster vaccination.** We next assessed the effect of a booster vaccination with MVA-MERS-S on humoral immunogenicity, hypothesizing that it may enhance the magnitude of the MERS-CoV-specific immune response. Two distinct IgG ELISAs specific for the S1-subunit of the MERS-CoV spike glycoprotein (MERS-CoV-S) were performed on B:D0, B:D7, B:D14, and B:D28, and included a control group of healthy, unvaccinated individuals with matching time-points (Fig. 3). All individuals who received the booster vaccination had no detectable antibody titers on B:D0 prior to vaccination, as measured in both the in-house (Fig. 3a) and the EUROIMMUN (Fig. 3b) ELISAs (in-house ELISA optical density (OD) 0.08 [95% confidence interval (CI) 0.03–0.13], cut-off 0.5, all EUROIMMUN ELISA results see Supplementary Data 1). After booster vaccination, antibody titers increased rapidly (by B:D7) and reached levels comparable to the maximum titers observed after the initial two vaccinations (OD exemplary for the in-house ELISA: 1.05 [95% CI 0.76–1.44] on D42 vs. 1.01 [95% CI 0.66–1.54] on B:D7, $p = 0.83$). By B:D14, titers further increased to significantly higher levels compared to D42 (OD exemplary for the in-house ELISA, Fig. 3a: 1.78 [95% CI 1.57–2.02] on B:D7, $p = 0.004$) and remained high on B:D28 in both assays (OD exemplary for the in-house ELISA: 1.91 [95% CI 1.73–2.11]). All individuals (10/10) seroconverted by B:D14. No anti-MERS-CoV-S1 antibody titers were detected in the serum of the controls ($n = 2$ in the in-house and $n = 4$ in the EUROIMMUN ELISA, respectively).

Two neutralization assays, namely the virus neutralization test (VNT) and the plaque reduction neutralization test assessing a $\geq 80\%$ reduction of plaques ($PRNT_{80}$), showed dynamics similar to the ELISA assays. A strong correlation between binding and neutralizing MVA-MERS-S-induced anti-MERS-CoV-specific antibody responses had previously been reported (measured via in-house ELISA and $PRNT_{80}$)[18] and we showed a comparably high inter-assay correlation between in-house-ELISA and VNT (Spearman $r = 0.78$ [95% CI 0.56–0.89], $p < 0.0001$; Supplementary Fig. 3a), as well as VNT and $PRNT_{80}$ (Spearman $r = 0.84$ [95% CI 0.67–0.92], $p < 0.0001$; Supplementary Fig. 3b). NAb levels increased early after booster vaccination and reached maximum levels by B:D14 and B:D28 in $PRNT_{80}$ (Fig. 4a) and VNT (Fig. 4b), respectively. These levels peaked earlier and exceeded the maximum levels observed after the initial vaccination regimen by more than ten-fold in both neutralization assays, specifically 11-fold in the $PRNT_{80}$ (reciprocal titer 49.2 [95% CI 30.8–78.8] on D35 vs. 520 [95% CI 346–782] on B:D14) and

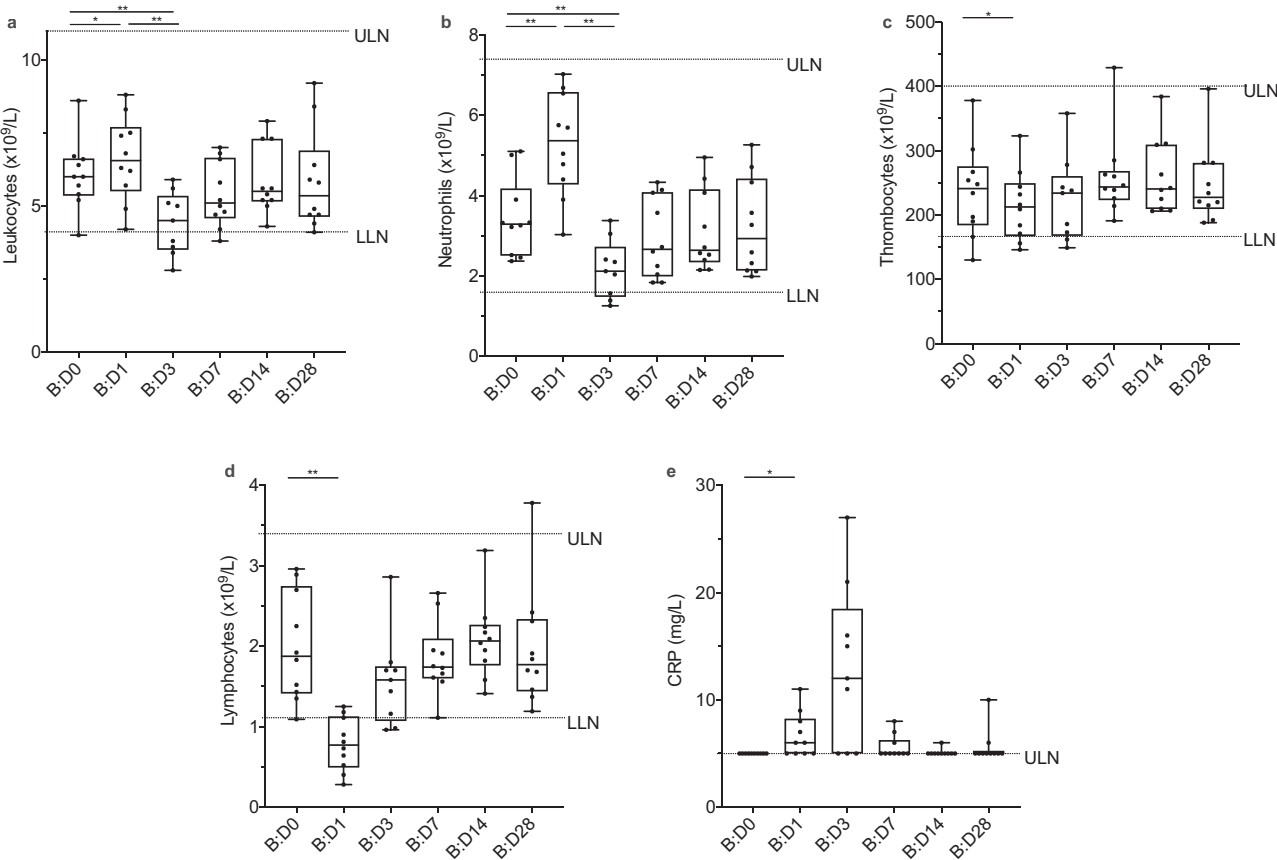

**Fig. 2 Biologic monitoring.** Graphs represent changes in **a** leukocyte, **b** neutrophil, **c** thrombocyte, and **d** lymphocyte counts, as well as **e** C-reactive protein (CRP) levels, in all booster study participants ($n = 10$) after booster vaccination. We observed an increase in **a** leukocyte ($p = 0.014$) and, by extension, **b** neutrophil counts ($p = 0.002$) as well as **e** CRP levels ($p = 0.031$) and a decrease in **c** thrombocyte ($p = 0.049$) and **d** lymphocyte ($p = 0.002$) counts on boost (B) Day 1 (B:D1) compared to B:D0. By B:D3, leukocyte and neutrophil counts had decreased ($p = 0.004$ for both neutrophil and leukocyte count on B:D1 vs B:D3 and B:D0. vs. BD3). These changes from baseline were transient and not clinically significant, but indicate biologic activity after vaccination. CRP levels of <5 mg/L (below the limit of detection) were set to 5 mg/L for visualization. Boxes indicate 25–75 percentile; whiskers are min. to max.; medians are shown as horizonal lines within the boxes. ULN = upper limit of normal. LLN = lower limit of normal. *$p < 0.05$, **$p < 0.005$, differences assessed using a two-sided Wilcoxon matched-pairs signed rank test. Source data are provided as a Source Data file.

12-fold in the VNT (reciprocal titer 5.10 [95% CI 1.74–14.9] on D42 vs. 60.7 [95% CI 39.0–94.3] on B:D28). Remarkably, all 5 participants who had not developed nAb levels above the threshold for positivity after the initial two vaccinations as measured by VNT had now developed positive nAb levels by B:D14 (Fig. 4c). Neutralizing antibodies against MERS-CoV were not detected in the serum of controls ($n = 4$; Fig. 4a, b).

**Characterization of immunogenic B cell epitopes**. To further characterize antibody generation to MERS-CoV-S-specific linear epitopes, we used microarrays mapping the proteome of MERS-CoV-S. Sera of all ten booster dose recipients and one unvaccinated control (see Supplementary Fig. 4) were screened for IgA, IgM, and IgG with 15-mer peptides spanning the whole MERS-CoV-S protein (two amino acids (AA) lateral shift, 670 peptides in total) on D0, D28, D42, B:D0, and B:D28. Figure 5 shows a schematic of MERS-CoV-S (Fig. 5a) with a heatmap of IgG binding to the individual MERS-CoV-S epitopes (Fig. 5b).

While we did not observe significant binding of IgA and IgM antibodies to MERS-CoV-S epitopes (Supplementary Data 2), we identified a significant induction of antibody responses to four immunodominant IgG epitopes on B:D28 as compared to baseline, namely AA sequences 535–553, 887–913, 1225–1247, and 1333–1353 on MERS-CoV-S (Fig. 5c–f).

AA 535–553, which consists of three overlapping peptides (OLP), is located in the immunogenic RBD of MERS-CoV-S (Fig. 5a). While we observed a trend towards an increase of AA 535–553-specific IgG by D42 and a decline by B:D0, significant epitope reactivity was only observed after booster vaccination (B:D28, Supplementary Fig. 5). Epitope 887–913 (Fig. 5d), which contains five immunogenic OLP (peptides AA 893–907 and AA 895–909 were excluded due to low antibody binding on B:D28), is located in the S2 subunit of MERS-CoV-S adjacent to the S2' cleavage site. Here, significant epitope binding was observed after primary vaccinations with the exception of peptide AA 891–905, for which binding was observed only after booster vaccination. Epitopes 1225–1247 (Figs. 5e) and 1333–1353 (Fig. 4f), consisting of five and four OLP respectively, are both located in the stem helix of the S2 subunit, with epitope 1333–1353 being located within the transmembrane region. For both epitopes, we observed a significant increase in serum levels of epitope-specific antibodies as early as D28 after prime vaccination, a further increase by D42, followed by a decline by B:D0 and a strong re-induction by B:D28 (Supplementary Fig. 6).

**Humoral cross-reactivity to other human coronaviruses**. It has been observed that MERS-CoV infection leads to an increase of antibody titers against other common cold HCoV[19] and MERS-CoV vaccine-induced cross-reactive antibodies to other HCoV

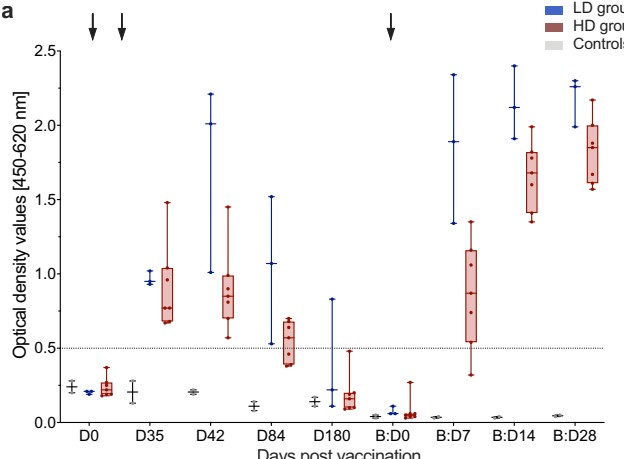

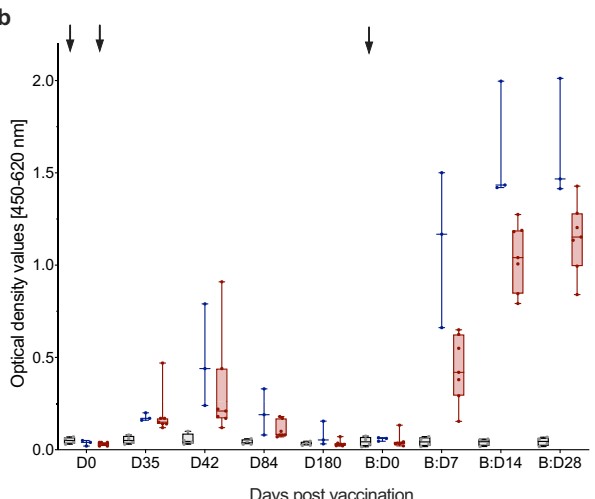

**Fig. 3 Increase in anti-MERS-CoV-S-specific binding antibodies after booster vaccination.** Anti-MERS-CoV-S-specific binding antibodies were measured as optical density values (y-axes) by two distinct ELISAs **a** in-house ELISA, **b** EUROIMMUN ELISA) at multiple timepoints after first (Day (D) 0 (D0), D35, D42, D84, D180) and booster (B) (B:D0, B:D7, B:D14, B:D28) vaccinations (x-axes). Former low dose (LD, n = 3) and high dose (HD, n = 7) vaccinees are depicted in blue and red, respectively, controls in gray (n = 2 in (**a**) and n = 4 in (**b**)). Boxes indicate 25–75 percentile; whiskers are min. to max.; medians are shown as horizonal lines within the boxes. Arrows indicate vaccinations. The horizontal dashed line in (**a**) indicates the cut-off level for positivity. Source data are provided as a Source Data file.

have been described in a mouse model[20]. To examine if prior exposure to other HCoV influences MVA-MERS-S-induced MERS-CoV-S-specific antibody generation, we assessed anti-S antibodies to HCoV-OC43, -229E, -HKU1, -NL63 (Fig. 6a–d) as well as MERS-CoV (Fig. 6e) and SARS-CoV (Fig. 6f) using a CoV spike protein-based immunofluorescence assay (IFA). We screened all vaccinees who had received the initial vaccination regimen with vaccinations on Days 0 and 28 (n = 23) and negative controls (n = 6) at baseline (D0) and 14 days after the second vaccination (D42). We did not find antibody reactivity against SARS-CoV spike protein, which served as the control antigen. Likewise, anti-MERS-CoV-S antibodies were not detected in controls and in D0 specimen from vaccinees, but in contrast, were significantly increased in LD as well as HD vaccinees on D42. Anti-MERS-CoV-S immunofluorescence titers on D42 strongly correlated with nAb as measured in PRNT$_{80}$

(Spearman r = 0.86 [95% CI 0.71–0.93], p < 0.0001; Supplementary Fig. 7), further cross-validating the individual assays. For the four common-cold HCoV, the majority of participants showed positive antibody titers at baseline (Supplementary Table 3). While the individual participants had variable titers, there was no correlation between baseline HCoV and anti-MERS-CoV-S titers on D42 (Supplementary Table 4) or fold-change of HCoV titers compared to MERS-CoV-S titers between pre (D0) and post vaccination (D42) timepoints (Supplementary Table 5).

## Discussion

We here investigated the safety and immunogenicity profile of an additional booster vaccination with the MVA-based vaccine candidate MVA-MERS-S against MERS-CoV one year after primary immunization in a Phase 1 clinical trial.

While the study was not powered for formal statistical analyses, the reactogenicity of the booster vaccination appeared to be comparable to the initial primary immunization regimen with regard to dynamics, type, and severity of AE. Biologic monitoring likewise demonstrated a good tolerability of the booster vaccination. We observed mild and transient changes in hematologic parameters and CRP. Transient changes, specifically of hematologic parameters, have been observed after vaccination—including in our initial study of two immunizations with MVA-MERS-S—and can be interpreted as a sign of biologic response to vaccination[18,21]. We found that a late homologous booster of MVA-MERS-S potently increased insert-specific immunity. Anti-MERS-CoV-specific binding and neutralizing antibody titers increased in all participants, irrespective of initial LD or HD primary immunizations, and even if they had previously failed to mount neutralizing antibodies following the original two-dose regimen. Of note, mean neutralizing antibody levels were up to 12-fold higher than the maximum levels observed after two vaccinations.

The licensed MVA vaccine against poxvirus is applied on Days 0 and 28. Consequently, this schedule has often been employed in the development of MVA-based vector vaccines[22]. Early secondary immunizations are chosen to maximize protective immunity in a timely manner, which is of particular importance in the context of outbreaks. However, additional and/or delayed booster vaccination may be required to increase the magnitude and duration of immunity and as a result, many established immunization regimens include a late booster vaccination in their regular schedule[23]. Recently, the question of whether, when and how many booster vaccinations may be necessary for optimal protection has gained increased attention and has been a topic of intense debate in the context of SARS-CoV-2 vaccine development. Clinical trials and observational studies of booster vaccinations with distinct COVID-19 vaccines revealed acceptable safety profiles, which were comparable to those of primary immunization[24]. Booster vaccinations against SARS-CoV-2, given at least 5 months after the primary series, increased immunogenicity and efficacy as well as the breadth of immune responses and level of cross-protection from symptomatic infection with variants[25,26]. A COVID-19 booster vaccination also substantially increased immunogenicity in immunocompromised and elderly individuals, who are at an increased risk for severe disease and may not mount sufficient and/or durable anti-SARS-CoV-2 immunity after primary immunization[27,28]. As a result, several national COVID-19 immunization guidances now recommend booster vaccinations 3–6 months after primary immunization[29,30].

With regard to MVA-vectored vaccines, previous studies likewise support the rationale of implementing a booster vaccination. In a clinical Phase 1/2a trial of an MVA-based H5N1

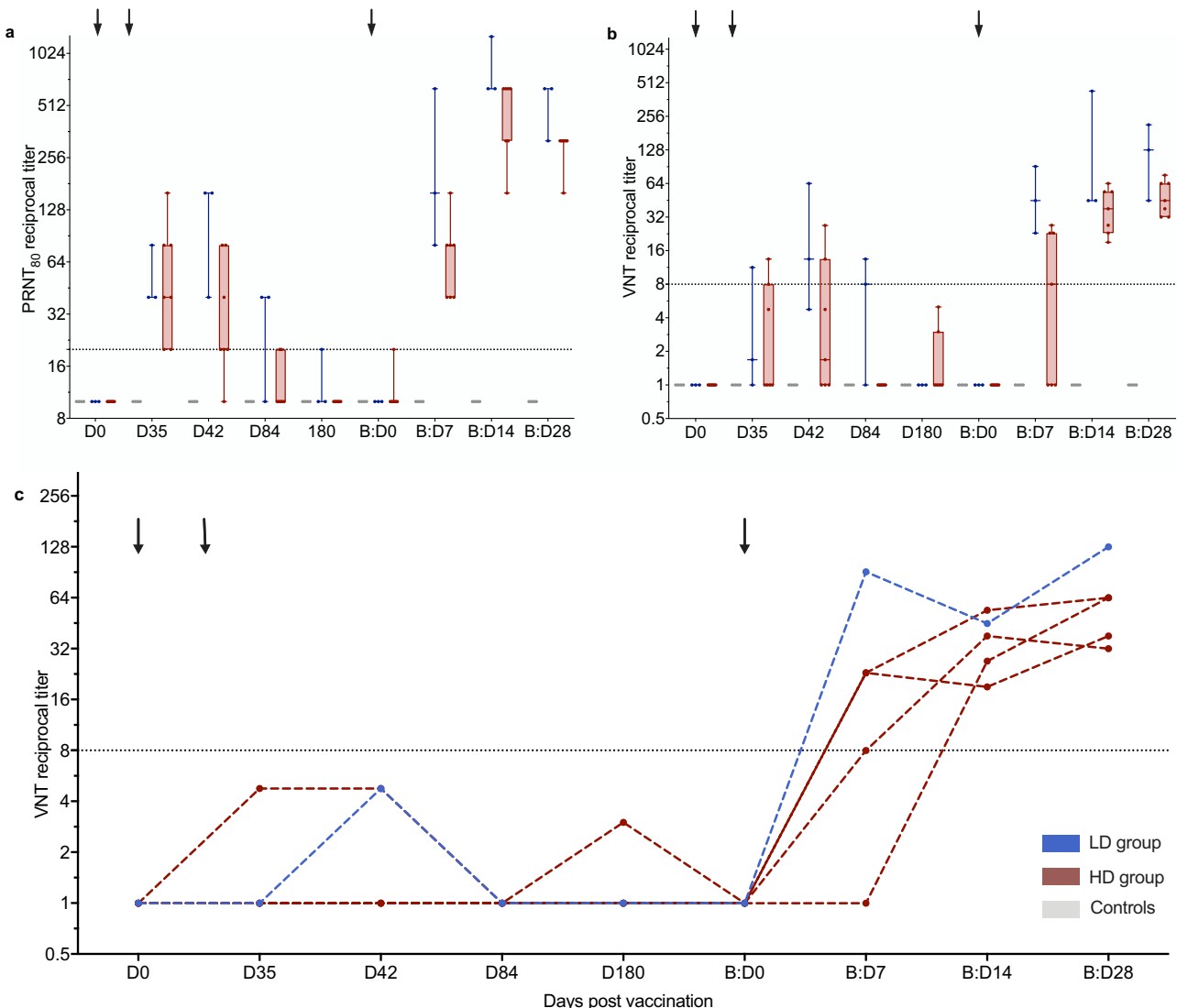

**Fig. 4 Increase in MERS-CoV neutralizing antibodies after booster vaccination.** MERS-CoV neutralization was measured by **a** ≥80% plaque reduction neutralization test (PRNT$_{80}$) and **b**, **c** virus neutralization test (VNT). Neutralizing antibody levels were measured in reciprocal titers (y-axes) at multiple timepoints after first (Day (D) 0 (D0), D35, D42, D84, and D180) and booster (B) (B:D0, B:D7, B:D14, and B:D28) vaccinations (x-axes). Arrows indicate vaccinations. Horizontal dashed lines indicate the cut-off levels for positivity. **a+b** Boxes indicate 25–75 percentile; whiskers are min. to max.; medians are shown as horizonal lines within the boxes. Former low dose (LD, $n = 3$) and high dose (HD, $n = 7$) vaccinees are depicted in blue and red, respectively, controls in gray ($n = 4$). **c** VNT titers after prime vaccination measured longitudinally in the five individuals who did not develop nAb after prime vaccinations. Source data are provided as a Source Data file.

influenza vaccine, a booster vaccination one year after an initial 28-day prime-prime regimen markedly increased binding and neutralizing antibody titers[31]. Booster vaccinations of MVA-vectored HIV vaccine candidates re-induced humoral and cellular insert-specific immunity when administered as late as 3–4 years after the last vaccination[32,33]. Furthermore, a 56-day interval between vaccinations led to an increased and more durable immune response compared to a 14-day interval in a non-human primate model of an MVA-based HIV vaccine, likely due to improved B-cell priming and innate immune responses[34]. These data further support the rationale of implementing booster vaccinations in immunization regimens with MVA-based vaccine candidates.

The identification of immunogenic epitopes is another essential aspect for vaccine development and assessment, and the generation of a broad humoral immune response is desirable. While the RBD of MERS-CoV-S represents an important antigenic target to induce neutralizing antibodies[35], MERS-CoV-S epitopes outside the RBD contribute further to vaccine- or virus-induced humoral immunity. In mice, MERS-CoV vaccine candidates containing the whole spike protein have been shown to induce superior immunity to vaccine candidates containing the RBD only, and this was also demonstrated for vaccine protection capacity in non-human primates[36]. Data from mice vaccinated with MVA-MERS-S show that the neutralizing capacity induced by the vaccine is only partially mediated by anti-RBD antibodies[37] and murine transfer studies of human monoclonal antibodies targeting non-RBD spike domains have demonstrated protection to lethal MERS-CoV challenge[38]. In addition, the induction of humoral immunity to multiple RBD and non-RBD antigenic sites on the spike glycoprotein can defer the development of escape mutations[39].

In this study, we identified four immunogenic linear MERS-CoV-S IgG epitopes induced by vaccination with MVA-MERS-S.

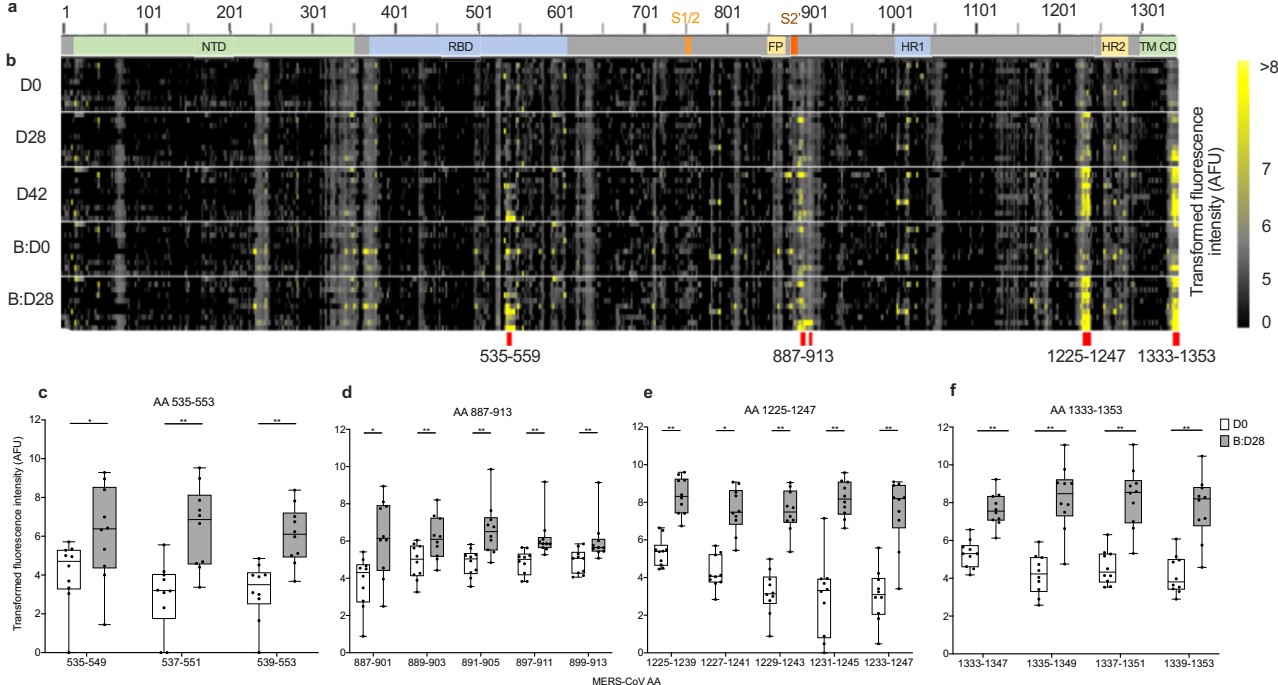

**Fig. 5 Identification of four immunogenic MERS-CoV-S epitopes. a** Schematic representation of the MERS-CoV-S protein. The N-terminal domain (NTD), receptor-binding domain (RBD), S1/S2 cleavage site (S1/S2), fusion peptide (FP), S2' cleavage site (S2'), heptad repeat 1 and 2 (HR1, HR2), transmembrane domain (TM) and cytoplasmic domain (CD) are illustrated. **b** Microarray of 15-mer peptides spanning the complete MERS-CoV-S protein with a 13 amino acid (AA) overlap. Immunogenic B-cell peptides are marked with red lines. **c–f** IgG binding to the respective peptides on MERS-CoV-S was measured in fluorescence intensity (as arbitrary fluorescence units, AFU), depicted as transformed values (areas sinus hyperbolicus (asinh), y axis), in all booster study participants ($n = 10$). Mean levels of peptide-binding IgG at baseline (Day (D) 0 (D0), white boxes) and 28 days after booster vaccination (Boost Day (B:D) 28 (B:D28), gray boxes) were compared using a two-sided Wilcoxon matched-pairs signed rank test and are depicted for each peptide (x-axis) within the immunogenic epitopes **c** AA 535–553 (AA 535–549, $p = 0.014$; AA 537–551, $p = 0.002$; AA 539–553, $p = 0.002$), **d** AA 887–913 (AA 887–901, $p = 0.027$; AA 889–903, $p = 0.006$; AA 891–905, $p = 0.002$; AA 897–911, $p = 0.002$; AA 899–913, $p = 0.002$), **e** AA 1225–1247 (AA 1225–1239, $p = 0.002$; AA 1227–1241, $p = 0.002$; AA 1229–1243, $p = 0.002$; AA 1231–1245, $p = 0.002$; AA 1233–1247, $p = 0.004$) and **f** AA 1333–1353 (AA 1333–1347, $p = 0.002$; AA 1335–1349, $p = 0.002$; AA 1337–1351, $p = 0.002$; AA 1339–1353, $p = 0.002$). Boxes indicate 25–75 percentile; whiskers are min. to max.; medians are shown as horizonal lines within the boxes. *$p < 0.05$, **$p < 0.005$. Source data are provided as a Source Data file.

While epitope AA 535–553 is located in the immunogenic RBD, 3/4 epitopes were located in the S2 subunit. Epitope AA 887–913 lies within the S2' cleavage site, a conserved region shown to play a role in the induction of neutralizing antibodies in SARS-CoV-2 infection[40]. We, furthermore, found two immunogenic epitopes, AA 1333–1353 and AA 1225–1247, located in the MERS-CoV-S stem helix. For three of the immunogenic epitopes we identified in our study, namely AA 887–913, AA 1225–1247, and AA 1333–1353, we observed an increase in antibody reactivity as early as D28, and antibody reactivity further increased after booster vaccination for individual peptides within these epitopes. In contrast, epitope AA 535–553 only showed reactivity after three vaccinations, indicating an increased breadth of immune response after the booster vaccination.

With regard to the two epitopes we identified within the stem helix, two human monoclonal anti-MERS-CoV antibodies (28D9 and 1.6C7) have recently been identified that both target a linear peptide in this location (AA 1229–1243)[20]. These antibodies could not only be induced by MERS-CoV-S immunization of mice, but were also present in convalescent sera of humans and dromedary camels after natural infection with MERS-CoV. Moreover, they effectively neutralized MERS-CoV and showed protective capacity in mice. In contrast, antibody reactivity was not observed in uninfected human or camel control sera[20]. These data suggest that the antibodies induced by MVA-MERS-S and specifically targeting epitope AA 1225–1247 may also play an important role in vaccine-conferred immunity. They further

support our observations that antibody binding to these epitopes was specific, as we did not find relevant binding to any of the four epitopes we describe prior to vaccination with MVA-MERS-S.

The S2 subunit is highly conserved among betacoronaviruses. Immune responses targeted towards epitopes conserved among different coronaviruses may confer cross-reactivity and could be of relevance in delivering cross-protection. A study assessing the prevalence of anti-SARS-CoV-2 antibodies in an unexposed population did detect anti-S2-specific SARS-CoV-2 antibodies— while not detecting RBD-specific responses—and the authors found a strong correlation of anti-HCoV-OC43 spike protein antibodies and anti-SARS-CoV-2-S2-antibodies in a COVID-19 convalescent cohort[41]. The monoclonal anti-S2 directed antibodies 28D9 and 1.6C7 likewise showed cross-binding to other human betacoronaviruses, and in particular to HCoV-OC43[20]. In this study, we aimed to understand whether prior exposure to HCoV had an influence on the magnitude of anti-whole MERS-CoV-S immune responses after vaccination with MVA-MERS-S. For the whole spike protein, however, we did neither find evidence that humoral immunity to HCoV influenced the generation of anti-MERS-CoV antibody responses, nor did we see cross-reactivity of HCoV antibodies to MERS-CoV and SARS-CoV.

Our results add valuable information for the advancement of MERS-CoV vaccine candidates and, more generally, the MVA vector vaccine platform, which is now also being assessed in the context of SARS-CoV-2 vaccine development. Recently, an MVA-vectored vaccine candidate expressing a pre-fusion stabilized

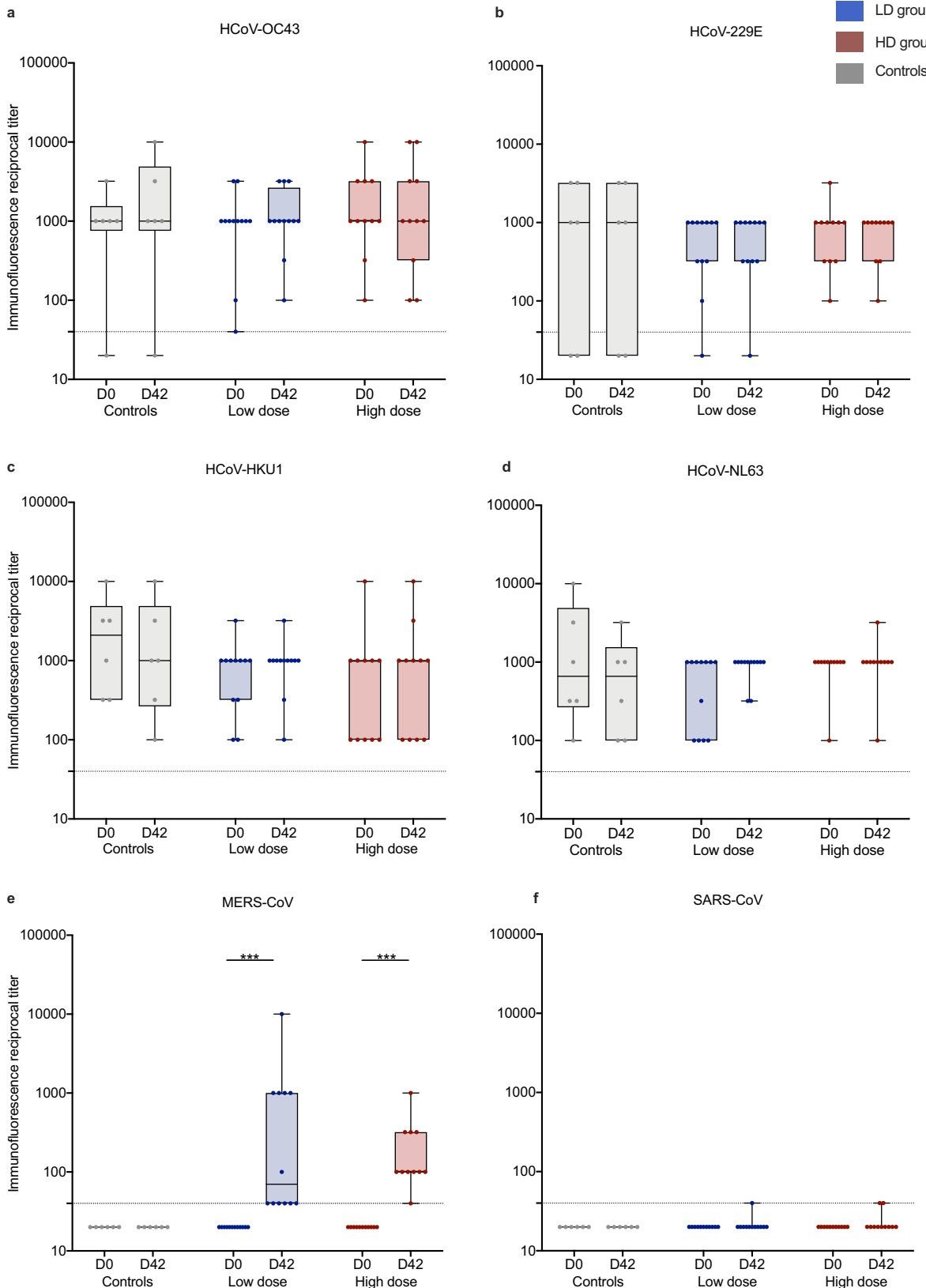

SARS-CoV-2-S has been shown to efficiently induce binding and neutralizing antibodies and reduce viral replication after challenge in a non-human primate model[42]. A further vaccine candidate, MVA-SARS-2-S, expressing full-length SARS-CoV-2 spike glycoprotein (NCT04569383), demonstrated immunogenicity and protective capacity in a mouse model[43] and we initiated a

Phase 1b clinical trial of a modified vaccine construct based on the same platform, MVA-SARS-2-ST, expressing the pre-fusion-stabilized spike glycoprotein (NCT04895449) at our research site in July 2021.

This study also has limitations. As an extension of the original Phase 1 trial of MVA-MERS-S, only 23 individuals were

**Fig. 6 Immunity to endemic HCoV compared to anti-MERS-CoV immunity.** Reciprocal anti-CoV spike protein antibody titers of **a** human (H) Coronavirus (CoV) (HCoV)-OC43, **b** HCoV-229E, **c** HCoV-HKU1, **d** HCoV-NL63, **e** MERS-CoV, and **f** SARS-CoV (used as control antigen) were measured on Day (D) 0 (D0, pre-vaccination) and D42 via immunofluorescence assay in all participants who completed the primary immunization schedule with MVA-MERS-S on D0 and D28 ($n = 23$) and unvaccinated controls ($n = 6$). Low dose ($n = 12$) and high dose ($n = 11$) vaccinees are depicted in blue and red, respectively, unvaccinated controls in gray. A titer of <40 (dotted line) was considered negative. While we observed increased anti-MERS-CoV titers on Day 42 compared to baseline, there was no significant increase of anti-HCoV titers. Boxes indicate 25–75 percentile; whiskers are min. to max.; medians are shown as horizonal lines within the boxes. **e** ***$p = 0.001$ (LD) and $p = 0.001$ (HD), assessed via two-sided Wilcoxon matched-pairs signed rank test. Source data are provided as a Source Data file.

potentially eligible to participate in this study. In addition, at the time of re-enrollment, only 10 individuals could accommodate the frequent and relatively lengthy study visits that were required for a detailed safety assessment of the booster vaccination (six study visits within 28 days including a dosing visit with a 6-hour observation period). Consequently, only a small and homogeneous group of healthy, Caucasian, female individuals could be enrolled in this study. Additionally, due to constraints in the study design, the booster vaccination was given in a relatively wide time window (12 ± 4 months). Concerning the assessment of immunogenic epitopes, we could, furthermore, not identify conformational or discontinuous B-cell epitopes since we were only able to assess linear epitopes via microarray. The findings can be interpreted as proof-of-concept and will need to be validated in future studies involving a bigger and more representative cohort.

We believe that the insights provided here lay the foundation for the further development of MVA-based vaccine candidates. A second generation MVA-MERS-S-DF1 vaccine candidate has now advanced to clinical development and is currently assessed in a double-blind, two-center, randomized, placebo-controlled Phase 1b trial (NCT04119440) supported by CEPI. This ongoing study builds on our data by using a larger cohort to evaluate different dose levels, distinct prime-prime intervals (28 vs. 56 days) and, specifically, a third vaccination one year after prime.

Taken together, we demonstrate in this study that a late booster vaccination with the vector vaccine candidate MVA-MERS-S against MERS-CoV is safe and significantly increases humoral immunogenicity including responses to four relevant IgG epitopes.

## Methods
**Study design and participants.** As a follow-up to the initial Phase 1 clinical trial of MVA-MERS-S, where a two-dose vaccine regimen was administered on Days 0 and 28[18], we designed a proof-of-concept study to assess the safety, tolerability, and immunogenicity of an additional third vaccination of MVA-MERS-S as a booster dose 12 ± 4 months after prime immunization. The primary objective was to assess the overall tolerability and safety of the MVA-MERS-S vaccine. Local and systemic reactogenicity were measured for 14 days following the booster vaccination using a diary, and adverse events, as well as changes in safety laboratory measures, were assessed at all study visits. The secondary objective was to evaluate MVA-MERS-S-specific antibody responses after MVA-MERS-S vaccination, which were assessed by measuring MERS-CoV-S-specific antibody levels via ELISA. An exploratory objective was the detailed characterization of vaccine-induced humoral responses. All data collection was performed in Hamburg, Germany: The clinical research organization Clinical Trial Center North in Hamburg performed the operative and regulatory project management; the University Medical Center Hamburg-Eppendorf was the Sponsor of this investigator-initiated trial. All study participants who had received both primary vaccinations (Days 0 and 28, $n = 23$) were invited to participate in this follow-up study via phone call. The full amended study protocol including all inclusion and exclusion criteria is provided in the Supplementary Information.

In addition, healthy unvaccinated individuals were recruited into an observational study and donated blood samples, which served as controls for immunogenicity assays.

**Study approval.** The amendment to the clinical trial was reviewed and approved by the Competent National Authority (Paul-Ehrlich-Institute) and the Ethics Committee of the Hamburg Medical Association (reference number PVN5531). The trial was conducted as an investigator-initiated trial under the sponsorship of

the University Medical Center Hamburg-Eppendorf (Hamburg, Germany) in accordance with ICH-GCP and is registered at ClinicalTrials.gov NCT03615911, EudraCT No. 2014-003195-23.

The observational study was registered with and approved by the Ethics Committee of the Hamburg Medical Association (reference number PV4780).

The clinical and observational studies were performed in accordance with the Declaration of Helsinki in its version of Fortaleza (2013). Written informed consent was obtained from all individuals prior to inclusion in the study.

**Study procedures.** Prior to booster vaccination (B:D0), baseline clinical parameters were recorded and blood for laboratory safety and immunogenicity analyses was collected. Participants were observed for 6 h post vaccination to assess immediate vaccine reactions. They returned to the study site for study visits on B:D1, B:D3, B:D7, B:D14, and B:D28 (End of Study), on which blood samples for immunogenicity and/or safety assessments were drawn and AEs were evaluated.

**Serum collection.** Whole blood from vaccinees was drawn into serum gel-monovettes on B:D0 prior to vaccination, B:D7, B:D14, and B:D28, and the same procedure applied to unvaccinated individuals at matching timepoints. Serum monovettes were transported at room temperature and processed within 8 h of blood collection. After centrifugation at $2500 \times g$, serum was aliquoted into cryotubes, immediately placed on dry ice and stored at $-80\,^{\circ}C$ until further usage.

**Safety assessments.** Safety was evaluated in accordance with the safety assessments of the initial two-vaccination regimen[18]. AEs were reported in electronic case report forms (eCRF, secuTrial database version 5.1.0.20®). Solicited and unsolicited AEs were classified on the basis of their severity (mild/moderate/severe), seriousness, relation to the vaccine (related vs. unrelated), and quality (local vs. systemic). Grading was perfomed by qualified study personnel according to the Common Terminology Criteria for Adverse Events (CTCAE) version 4.0[44] and the U.S. Food and Drug Administration (FDA) Voluntary Guidance 2007[45]. AEs were classified as solicited if they occurred up to 14 days post vaccination and included local reactogenicity, gastrointestinal symptoms, fatigue/malaise, myalgia, arthralgia, headaches, and fever/chills (see study protocol in the Supplementary Information). Laboratory analyses included measurements of electrolytes, complete blood cell counts, creatinine, liver enzymes, CRP and troponin levels. AE, clinical, and biologic monitoring were performed longitudinally (Fig. 1).

**Binding anti-MERS-CoV-S1 antibody responses: EUROIMMUN MERS-CoV-S1 ELISA.** Two IgG MERS-CoV-S1-specific ELISAs were performed by two independent laboratories as conducted in the initial MVA-MERS-S clinical trial[18]. A commercial ELISA (EUROIMMUN AG, Lübeck, Germany, order number EI 2604-9601G) was performed according to the manufacturer's protocol. Sera (assessed in duplicates) were diluted 1:101 in sample buffer. The dilution was incubated with MERS-CoV-S1 antigen for 30 min and subsequently washed three times with washing buffer. Peroxidase-labeled anti-human-IgG was added and another 30 min incubation step followed. After another wash, substrate solution was added to each well. After 15 min of incubation, the reaction was stopped and the OD was measured at 450 nm. The samples were analyzed using the BEP® (Behring ELISA Prozessor) III system (Behring, Marburg, Germany). The cut off for positivity of each sample was calculated according to the manufacturer's instructions and a ratio of >1.1 (sample/calibrator) was considered positive.

**Binding anti-MERS-CoV-S1 antibody responses: in-house MERS-CoV-S1 ELISA.** The second ELISA utilized in this and the core study[18] was an in-house ELISA we performed using a previously established and validated protocol published by Okba et al.[46]. The sensitivities and specificities of both ELISA assays have previously been comparatively assessed and published[46]. High binding 96-well microtiter plates were coated overnight at 4 °C with 1 µg/ml MERS-CoV S1 protein. The plates were blocked for 1 h at 37 °C with 100 µl Blotto blocking buffer in tris-buffered saline (TBS) (ThermoFisher Scientific, Waltham, MA, United States) per well. Plates were washed three times with 100 µl/well phosphate-buffered saline (PBS) with 0.05% Tween-20. Sera were diluted in blocking buffer at a ratio of 1:100 and 100 µl/well of the dilution was added to the plates, incubated at 37 °C for 1 h, and washed another three times. 100 µl/well of horseradish peroxidase (HRP)

conjugated rabbit anti-human IgG (stock 1.3 g/L, Dako, Santa Clara, CA, USA), diluted 1:6000 in PBS were added and the plate was incubated for 1 h at 37 °C. After three washes, 100 µl/well 3,3′,5,5′-Tetramethylbenzidine (TMB) substrate was added and the plate was incubated for 5 min at room temperature. The reaction was stopped by adding 100 µl/well of 0.25 M sulfuric acid. Photometry was performed using a microplate reader (Tecan Infinite F200, Tecan, Männedorf, Switzerland) at a measurement wavelength of 450 nm with a reference wavelength of 620 nm. Results were reported as optical density (OD) values of the measurement wavelength, subtracted by the reference wavelength. The cut off was set at an OD of 0.5, which was >6 standard deviations (SD) of the negative cohort used to establish the assay.

**Anti-MERS-CoV neutralizing antibodies**. Neutralizing anti-MERS-CoV antibodies were analyzed via VNT and $PRNT_{80}$ using the same protocols as in the initial trial[18]. Briefly, the cytopathic effect (in the VNT) and plaque reduction of MERS-CoV-infected cells (in the PRNT) were measured using HuH-7 cells incubated with MERS-CoV isolate EMC/2012 (JCRB0403, JCRB cell bank of Okayama University) and serial dilutions of vaccinees' sera. For the VNT, complement inactivation of serum samples was carried out at 56 °C for 30 min. Afterwards, sera and a human monoclonal antibody (m336, Detai Bio-Tech Co., Nanjing, China), which served as a control for neutralization[35], were serially diluted in 96 well plates, starting from a 1:8 and a 1:16 (stock 100 µg/ml) dilution, respectively. Sera and the control were then incubated for 1 h at 37 °C together with MERS-CoV (EMC/2012 isolate; 100 50% tissue culture infective doses (TCID50)) in HuH-7 cells. Cytopathic effect (CPE) was analyzed four days post infection. Neutralization was defined as absence of CPE compared to virus controls. Neutralization titers were calculated as reciprocal values of geometric mean titers of four serum replicates and a titer of 1:8 was considered a positive response.

For the $PRNT_{80}$, sera were heat-inactivated and serially diluted starting at a diliution of 1:10. The samples were then mixed at a 1:1 ratio with 400 PFU of MERS-CoV (EMC/2012 isolate) and incubated for 1 h. The mix was then added on top of 96-well plates which were coated with a HuH-7 cell monolayer. After an incubation of 1 h, the mix was removed. The 96-well plates were incubated for another 8 h and then fixed and stained using anti-MERS-CoV-N protein mouse monoclonal antibody (Sino Biological, Eschborn, Germany) After the addition of a secondary peroxidase-labeled goat anti-mouse IgG1 (Southern Biotech) antibody, TMB (True Blue, KPL) was added to develop a signal. The number of infected cells per well was counted using the ImmunoSpot® Image analyzer (CTL Europe GmbH, Bonn, Germany). Neutralization levels were defined as the reciprocal values of a ≥80% reduction of plaques ($PRNT_{80}$) and a titer of ≥20 was considered positive (one measurement per sample).

**Peptide microarrays**. To assess antigenic peptides of MERS-CoV-S and identify B cell epitopes, we screened high-density peptide microarrays with sera of vaccinees using a protocol published by Heidepriem et al.[47]. The whole proteome of MERS-CoV-S (GenBank ID: AFS88936.1) consisting of 1353 AA was mapped as 670 15-mer peptides with a lateral shift of two AA as spot duplicates on a total of 55 peptide microarray replica, obtained from PEPperPRINT GmbH (Heidelberg, Germany). 55 sera (10 vaccinees and 1 control with 5 timepoints) were diluted 1:200 and incubated on the arrays overnight. IgA, IgM and IgG serum antibody interactions were detected with Fc-specific differentially labeled fluorescent secondary antibodies, namely 0.5 mg/ml anti-human IgG-Fc fragment (cross-adsorbed) DyLight 680 (A80-304D6, Bethyl Laboratories, Montgomery, TX, USA), 1.0 mg/ml anti-human IgM (mu chain) DyLight 549 (609-142-007, Rockland Immunochemicals Inc., Limerick, PA, USA), 1.0 mg/ml anti-human IgA (alpha chain) antibody DyLight 800 (609-145-006, Rockland Immunochemicals Inc., Limerick, PA, USA) and diluted in buffer (1:1000 for both anti-human IgG and IgA, and 1:2000 for anti-IgM). Fluorescence image acquisition was performed with the fluorescence scanner Genepix 4000B (Molecular Devices, San José, CA, USA) with a resolution of 5 µm, a photo multiplier gain of 600 and a laser power of 33%, and Odyssey Scanner (LI-COR Biotechnology Inc., Lincoln, NE, USA) with a resolution of 21 µm and a photo multiplier gain of 7. Image data was analyzed using PepSlide Analyzer software (version 1.5.8, SICASYS Software GmbH, Heidelberg, Germany) with the fixed-spot detection method, a spot width of 250 µm and spot height of 400 µm. The printed region was 190 µm × 338 µm. The median spot intensity was acquired for each spot. Data cleaning and aggregation of the raw data was performed with python 3 (version 3.7.1) using the numpy, pandas, re (regular expression), csv, and openpyxl packages and consequently exported to Excel. Antibody binding was measured as fluorescence binding intensity in arbitrary fluorescence units (AFU), which were transformed with the inverse hyperbolic function (area sinus hyperbolicus) for statistical analysis. A mean transformed fluorescence intensity of >6 on B:D28 was considered significant binding. Epitope binding with mean values <6 AFU on B:D28 were considered unspecific binding and the epitope was, therefore, excluded from statistical analysis.

**Differential HCoV and MERS-CoV IFA**. Presence of antibodies to HCoV-OC43, HCoV-229E, HCoV-HKU1, HCoV-NL63, and MERS-CoV was evaluated by recombinant CoV spike protein-based recombinant IFAs, basically following the protocol previously published by Corman et al.[48]. Vero B4 cells (DSMZ #ACC33; $10^6$ cells per ml) were transfected with respective spike-encoding pCG1-based expression plasmids using 2.5 µg of plasmid DNA and the FuGENE HD protocol (Roche, Basel, Switzerland). After an overnight incubation at 37 °C, the transfected cells were mixed at a 1:1 ratio with untransfected Vero B4 cells, spotted on glass slides, and incubated for an additional 6 h allowing cells to settle. Subsequently, cell fixation and permeabilization was achieved by immersing the slides into an ice-cold acetone/methanol mix (1:1 ratio). Blocking of unspecific binding was achieved by using 5% nonfat dry milk in PBS with 0.2% Tween for 1 h, followed by washing with PBS. Serial dilutions of vaccinee or control sera, diluted with sample buffer (EUROIMMUN AG, Lübeck, Germany), were applied in 30 µl, starting at a dilution of 1:40. Slides were incubated at room temperature for 1 h, washed with 0.02% Tween-containing PBS, and incubated for another 30 min at room temperature with 25 µl AlexaFluor488-labeled goat anti-human IgG (Jackson, Baltimore, PA, USA; affinity purified IgG (H+L); 1:200 of a 1 mg/ml stock). After another three washes, the slides were rinsed with water and mounted in DAPI ProLong mounting medium (Life Technologies). The samples were analyzed by using an Olympus microscope (BX60) at 200-fold magnification. Trained personnel performed two independent evaluations of fluorescence intensities. A reciprocal titer of ≥40 was considered the cut-off for positivity. For statistical calculations and visualization, reciprocal titers <40 (below the limit of detection) were set at a value of 20.

**Statistics**. Descriptive statistics were used to summarize metric (number, mean with SD, minimum, maximum, and median with IQR) and categorical data (frequencies). For OD values, geometric means along with two-sided 95% CI were calculated using logarithmic transformations. To avoid overinterpretation of data, a non-normal distribution was assumed. Wilcoxon matched-pairs signed rank test and Mann–Whitney-U test were used to analyze paired and unpaired samples, respectively. Correlations were calculated using Spearman r. Statistical calculation of the p-value was two-sided and a p-value of ≤0.05 was considered statistically significant. Statistical analysis was performed using GraphPad Prism version 8.4.2. Figures were created with GraphPad Prism version 8.4.2 and BioRender.com.

**Reporting summary**. Further information on research design is available in the Nature Research Reporting Summary linked to this article.

## Data availability

The data used in this study are provided as de-identified participant data in the Supplementary Information or in the Source Data file. The study protocol is available in the Supplementary Information of this manuscript. Source data are provided with this paper.

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

## Acknowledgements

We thank all study volunteers without whom this study would not have been possible. We thank Dr. Saskia Borregaard, Dr. Alen Jambrecina, and Laura Kaltenberg at the CTC North GmbH & Co. KG in Hamburg, Germany for regulatory support in submitting the study documents and for the operative management of the study conduct, as well as Elisabeth Möncke-Buchner and Petra Mackeldanz (both Charité-Universitätsmedizin Berlin, Institute of Virology) for excellent technical assistance. We are also grateful for the critical reading of the manuscript and the valuable advice provided by Dr. Sibylle Mellinghoff (Cologne University Hospital, Division of Infectious Diseases) and Elaine Hussey (University Medical Center Hamburg-Eppendorf, First Department of Medicine). This research was supported by the German Center for Infection Research (DZIF), grant numbers FKZ8009801908 (MMA), FKZ8009701702 (MMA), FKZ8033801809 (section emergency vaccines, S.B., V.K.), and FKZ8033701711 (infrastructure animal facility, S.B., A.K., V.K.). A.F. received a DZIF Clinical Leave Grant, grant number FKZ80095CLANF. C. Drosten was supported by the German Federal Ministry of Education and Research (BMBF) BMBF-RAPID 01KI1723A. V.M.C. was supported by BMBF project VARIPath (01KI2021). F.F.L. was supported by the BMBF grant 13XP5050A and the MPG-FhG cooperation grant Glyco3Display. B.L.H. is an inventor on a MERS patent with Erasmus MC Rotterdam (No. WO2014045254 A3, no payments to author or institution). The funder of this publicly-funded investigator-initiated trial had no role in study design, data collection, data analysis, data interpretation, or writing of the report. The corresponding author had full access to all the data in the study and had final responsibility for the decision to submit for publication.

## Author contributions

Conceptualization: M.M.A., S.B., T.H., and G.S.; methodology & supervision: M.M.A., S.B., C. Drosten, B.L.H., F.F.L., and G.S.; investigation: V.M.C., C. Dahlke, A.F., R.F., M.F., J.H., G.K., M.K., V.K., A.K., S.L., M.L.L., M.A.M., N.M.A.O., G.P., M.P.R., A.T., and A.V.; study conduct: M.M.A., A.F., T.K., and S.S.; formal analysis: C. Dahlke, A.F., M.K., V.K., A.K., S.L., F.F.L., M.A.M, N.M.A.O., and A.V.; writing – original draft: A.F.; writing – review and editing: M.M.A., C. Dahlke, B.L.H., T.H., G.K., T.K., V.K., F.F.L., M.A.M., G.S., and A.V.; visualization: C. Dahlke, A.F., S.L., and F.F.L.; project administration: M.M.A., C. Dahlke, and A.F.

## Funding

## Competing interests

The authors declare no competing interests.

## Additional information

[1]University Medical Center Hamburg-Eppendorf, Institute for Infection Research and Vaccine Development (IIRVD), Hamburg, Germany. [2]Bernhard-Nocht-Institute for Tropical Medicine, Department for Clinical Immunology of Infectious Diseases, Hamburg, Germany. [3]German Center for Infection Research, partner site Hamburg-Lübeck-Borstel-Riems, Hamburg, Germany. [4]University Medical Center Hamburg-Eppendorf, First Department of Medicine, Division of Infectious Diseases, Hamburg, Germany. [5]Philipps University Marburg, Institute of Virology, Marburg, Germany. [6]German Center for Infection Research, partner site Gießen-Marburg-Langen, Marburg, Germany. [7]Erasmus Medical Center, Department of Viroscience, Rotterdam, the Netherlands. [8]Max Planck Institute of Colloids and Interfaces, Department of Biomolecular Systems, Potsdam, Germany. [9]Charité-Universitätsmedizin Berlin, corporate member of Freie Universität Berlin, Humboldt-Universität zu Berlin, and Berlin Institute of Health, Institute of Virology, Berlin, Germany. [10]German Center for Infection Research, partner site Berlin, Berlin, Germany. [11]University of Veterinary Medicine Hanover, Institute of Virology, Hanover, Germany. [12]German Center for Infection Research, partner site Hanover-Brunswick, Hanover, Germany. [13]LMU University of Munich, Institute of Infectious Diseases and Zoonoses, Munich, Germany. [14]German Center for Infection Research, partner site Munich, Munich, Germany. [15]German Center for Infection Research, Translational Project Management Office, Brunswick, Germany. ✉email: m.addo@uke.de

