## [Peer Review File · Nature Communications]

Increased neutralization and IgG epitope identification after MVA-MERS-S booster vaccination against Middle East respiratory syndromeEditorial Note: Parts of this Peer Review File have been redacted as indicated to maintain the confidentiality of unpublished data.

Reviewers' Comments:

Reviewer #1:

Remarks to the Author:

Fathi et al present a manuscript detailed safety and immune responses of homologous boost vaccination with MVA-MERS. A total of 10 subjects received high-dose booster vaccination approximately 1 year post-primary series.

Overall, the paper is very well written and presented. Especially in the midst of the SARS-CoV-2 pandemic, the information presented represents critical advancements into the understanding of coronavirus vaccine-induced immune responses. The paper also provides critical information regarding the safety of MVA-vectored vaccines when used as a homologous boost.

The correlation of immune responses between in-house assays and commercial kits, as well as the B cell epitopes that are targeted are key advancements.

Comments and critiques are relatively minor.

1. Study design:

Recruitment: it is not stated either in the methods or in the results whether all subjects from the initial study (Koch et al Lancet) were contacted for participation in this study. The selection process, and whether some or a large fraction of the initial study declined participation (and why) should be detailed.

2. Safety:

2a. Transient changes in multiple hematologic parameters (platelets, neutrophils, lymphocytes) were noted in both the primary study as well as following booster vaccination. The authors should specifically comment on the relative change in these parameters and whether booster vaccination was associated with greater shifts, especially, for the subset of 10 individuals who participated in both studies.

2b. Do the authors consider the increase in neutrophils on Day 1 of any clinical significance - since this may be greater than that observed for the high-dose group with their first vaccination as part of the primary series.

2c. Were AEs and/or changes in hematologic parameters consistent between studies, or more random?

3. Epitope reactivity

It appears visually, based on examination of the microarray analysis in Fig 5 that reactivity to the three immunodominant epitopes correlated between the primary series and post-boost. In other words, there did not appear to be a significant maturation over time for those not reactive initially. Can the authors add comment as to differences / similarities of immune specificity in both periods.

4. T cell responses

T cell responses were investigated in the primary paper, but are not addressed in this study. This omission should be addressed; would be helpful to include this data.

Reviewer #2:

Remarks to the Author:

I congratulate the authors on presenting compelling data to support the role of a booster dose of this MERS-CoV vaccine candidate in eliciting immunogenicity.

The strengths of this study include the detailed analysis of humoral immunity to the level of individual epitopes using a peptide microarray. An additional potential strength would be the inclusion of long-term antibody persistence data following vaccine booster dose, although this data is referenced as

part of a separate manuscript under review.

The weaknesses of this study include small sample size of study participants and the absence of analysis of cellular immunity. The number of controls used is low; a larger number of controls in the antibody assays, ideally including some individuals who were vaccinated against SARS-CoV-2, would be useful to assess for low-level background antibody reactivity. In addition, the inclusion of control sera in the peptide microarray experiment would be useful to assess cross-reactivity of antibody responses to individual epitopes, which has particularly been observed for S2 epitopes. If possible, obtaining positive control sera from either human or dromedary camel cases of MERS-CoV would be useful to compare magnitudes of antibody responses to natural infection.

**Detailed point-by-point response to reviewer comments
Manuscript No.: NCOMMS-22-05187**

Reviewer comments

Reviewer #1:

Comment: Fathi et al present a manuscript detailed safety and immune responses of homologous boost vaccination with MVA-MERS. A total of 10 subjects received high-dose booster vaccination approximately 1 year post-primary series.

Overall, the paper is very well written and presented. Especially in the midst of the SARS-CoV-2 pandemic, the information presented represents critical advancements into the understanding of coronavirus vaccine-induced immune responses. The paper also provides critical information regarding the safety of MVA-vectored vaccines when used as a homologous boost.

The correlation of immune responses between in-house assays and commercial kits, as well as the B cell epitopes that are targeted are key advancements.

Comments and critiques are relatively minor.

Reply: We thank the reviewer for his/her valuable comments and the overall positive evaluation of the manuscript. We have addressed each of the comments and modified the manuscript to reflect the suggestions.

1. Study design:

Comment: Recruitment: it is not stated either in the methods or in the results whether all subjects from the initial study (Koch et al Lancet) were contacted for participation in this study. The selection process, and whether some or a large fraction of the initial study declined participation (and why) should be detailed.

Reply: We agree that providing more details on the recruitment will facilitate the understanding of the process and thank the reviewer for addressing this point. Briefly, all subjects were contacted by the study team via phone call and all subjects who were willing to participate in the study extension were invited to the study site to be re-screened. However, due to the necessity to attend multiple study visits within a small time window (28 days), only 10/23 individuals from the original trial were able to participate in the extension trial. We have now added this information in the results, discussion and method sections (lines 94-97, 335-339 and 363, respectively).

2. Safety:

Comment: 2a. Transient changes in multiple hematologic parameters (platelets, neutrophils, lymphocytes) were noted in both the primary study as well as following booster vaccination. The authors should specifically comment on the relative change in these parameters and whether booster vaccination was associated with greater shifts, especially, for the subset of 10 individuals who participated in both studies.

Reply: As the reviewer points out, we were able to closely monitor hematology and clinical chemistry parameters due to the frequent and early study visits after each vaccination. We now included an analysis of the relative changes after all vaccinations as Supplementary Figure 2 in the three days after vaccinations, when these transient changes occurred. Since 3/10 individuals had received a lower dose during the primary trial (1×10^7 PFU MVA-MERS-S) and 7/10 individuals had received the same dose initially (1×10^8 PFU MVA-MERS-S), we analyzed these two groups separately. As the number in each group is, therefore, small, we were careful not to overinterpret the analyses but rather included a descriptive statement in the results section (lines 139-144) and further elaborate in the discussion (lines 233-237). Please note that there was no study day 3 after the second vaccination in the initial trial.

Comment: 2b. Do the authors consider the increase in neutrophils on Day 1 of any clinical significance - since this may be greater than that observed for the high-dose group with their first vaccination as part of the primary series.

Reply: The increase of neutrophils was indeed higher after booster vaccination than after the first two vaccinations. We have now depicted this relative change in Supplementary Figure 2. We

observed that this increase was correlated with a decrease in lymphocyte counts, which we simultaneously observed, but not with CRP levels. We interpret these changes as a physiologic reaction to the vaccination, that may be increased compared to primary vaccinations and conforms with the higher humoral immunogenicity of the booster vaccination that we observed. Neutrophil counts on Day 1 after booster vaccination (B:D1), however, stayed within the physiologic range in all vaccinees (Fig. 2) and we, therefore, do not consider the increase of neutrophils as clinically significant (now described in the results in lines 144-147).

Comment: 2c. Were AEs and/or changes in hematologic parameters consistent between studies, or more random?

Reply: This is an interesting question. While this trial was not powered to run a formal statistical comparison of AE between the vaccinations, we now have included Supplementary Figure 1, depicting the solicited local and systemic AE after each vaccination. For reasons of comparability, we only included the 7 vaccinees who had received 1×10^8 PFU MVA-MERS-S as both the prime and booster doses. We describe this in the results section (lines 118-126). We also discuss the relative changes of hematologic parameters (see replies to comments 2a and 2b).

3. Epitope reactivity

Comment: It appears visually, based on examination of the microarray analysis in Fig 5 that reactivity to the three immunodominant epitopes correlated between the primary series and post-boost. In other words, there did not appear to be a significant maturation over time for those not reactive initially. Can the authors add comment as to differences / similarities of immune specificity in both periods.

Reply: We thank the reviewer for raising this point. Increased breadth of antibody responses to multiple antigens can often be observed in natural infection. Here, the immune system is typically facing large epitope diversity, and as a result, both a broader antibody response as well as the development of new and loss of initially observed epitope-reactivity may be seen after multiple exposures with the same pathogen. Crompton et al. have, for example, examined this aspect in plasmodium falciparum (Pf) infection, which is the causative agent for malaria tropica.¹ The authors used a protein microarray spanning around 23% of all 5,400 proteins of the extremely large Pf proteome to assess antibody reactivities in longitudinal human samples over an 8-month period. While they found that antibody reactivity and breadth generally increased with the number of exposures (i.e., both with age of the participants living in Sub-Saharan Africa, as well as later during the malaria season), the increase of antibody responses after the malaria season seemed to be short lived, when compared to subsequent levels in the dry season.

In contrast to natural infection, only very limited epitopes are expressed in vaccination. MVA-MERS-S only codes for one single protein, namely the spike glycoprotein, and the breadth of immune response observed in natural infection may therefore not be observed in vaccination. In our study, we identified four immunodominant epitopes. For three of these epitopes, namely AA 887-913, AA 1,225-1,247 and AA 1,333-1,353, we observed an increase in antibody reactivity as early as D28. However, antibody reactivity further increased by B:D28 as compared to peak values at D42 for individual peptides within these epitopes, i.e., AA 891-905, EDLLFDKVTIADPGY,

AA 897-911 KVTIADPGYMQGYDD and AA 1225-1239, NSTGIDFQDELDEFF, see also Supplementary Figure 6. In contrast, epitope AA 535-553 only showed reactivity after three vaccinations, indicating an increased breadth of immune response following the booster vaccination (see Supplementary Figure 5). We have included this comment in the manuscript in (lines 193-199 and 297-301).

4. T cell responses

Comment: T cell responses were investigated in the primary paper, but are not addressed in this study. This omission should be addressed; would be helpful to include this data.

Reply: We agree that cellular immunity needs to be addressed in the context of the booster vaccination extension trial and compared to the magnitude of T cell responses observed in the initial trial. In the submitted manuscript, we decided to focus on the primary and secondary objectives of the clinical trial, namely the safety and elicited antibody responses of MVA-MERS-S, with focus on a detailed investigation of humoral responses. We, however, also analyzed cellular immunity using an Interferon- γ ELISpot assay as described in the primary paper² and report the results in a separate manuscript currently under review (Weskamm et al.), which we have provided during the submission of this paper for the reviewers' reference. We have now also provided these results more specifically in the manuscript (lines 248-251). Specifically, we did observe a re-induction of T cell responses following booster vaccination, however, in contrast to the antibody responses, cellular immunity did not surpass the initial T cell frequencies after booster vaccination compared to the primary vaccination regimen. Figure 1, taken from Weskamm et al., illustrates this finding.

[Redacted]

Reviewer #2:

Comment: I congratulate the authors on presenting compelling data to support the role of a booster dose of this MERS-CoV vaccine candidate in eliciting immunogenicity. The strengths of this study include the detailed analysis of humoral immunity to the level of individual epitopes using a peptide microarray.

Reply: We would like to thank the reviewer for the positive feedback and for providing us with the opportunity to add further detail to the points raised by him/her.

Comment 1: An additional potential strength would be the inclusion of long-term antibody persistence data following vaccine booster dose, although this data is referenced as part of a separate manuscript under review.

Reply: We agree that the assessment of long-term antibody persistence is of high interest. In the work currently under review, we assessed the persistence of MERS-CoV-S-specific B cells and IgG antibody subclass titers three years after primary vaccination. We found that high IgG antibody responses persisted throughout the follow-up period in all vaccinees. Figure 2, taken from Weskamm et al., illustrates this finding. For the reviewers' reference, we have provided the separate manuscript currently under review with the submission of this manuscript. We also discuss the findings in the discussion.

[Redacted]

Comment: The weaknesses of this study include small sample size of study participants and the absence of analysis of cellular immunity. The number of controls used is low; a larger number of controls in the antibody assays, ideally including some individuals who were vaccinated against SARS-CoV-2, would be useful to assess for low-level background antibody reactivity.

Reply: We acknowledge that the sample size of the study participants in this proof-of-principle trial is small. This small study size is due to the nature of this first-in-human Phase 1 clinical trial and the respective study extension, which included individuals who were willing and able to continue the trial to receive a booster vaccination. We have addressed this limitation in the discussion (lines 335-341) and agree that a larger confirmatory trial will be necessary to validate the findings presented in this manuscript. To this end, we are currently conducting a two-center randomized-controlled Phase 1b trial of the next-generation MVA-MERS-S-DF1 vaccine candidate (manufactured on a different cell line) with a total of 160 participants (NCT04119440). This blinded trial is ongoing and also includes the assessment of a third (booster) vaccination to further validate the findings observed in the present smaller pilot study.

We have analyzed cellular immunity in the context of the booster vaccination and discuss this point under reviewer comment 4, “T cell responses”.

With regard to the controls in antibody assays, we have included two kinds of controls. Serum samples drawn from vaccinees prior to vaccination (baseline, Day 0, “D0”) served as intra-subject controls and were analyzed for each subject in each assay. No binding or neutralizing anti-MERS-CoV antibodies could be detected in any of the vaccinees at baseline. In addition, we obtained control sera from individuals who had not been vaccinated. The sera were drawn and processed at the same timepoints as those of the vaccinees. The in-house ELISA included matched sera of two, the commercial ELISA and both neutralization assays included matched sera of four control subjects for each timepoint.

The sensitivity and specificity of both in-house and the EUROIMMUN ELISA as well as the PRNT have previously been assessed by Okba et al. using serum samples of human MERS-CoV convalescent individuals with mild to severe disease and differing MERS-CoV antibody levels, healthy blood donors, as well as individuals with PCR-confirmed acute to convalescent non-MERS-CoV human coronavirus infections.³ The authors demonstrated a high specificity for all assays (99-100%) and a varying sensitivity, with the in-house ELISA showing the highest sensitivity (100%). We have now referenced this paper in the methods section and also included a statement about the sensitivity and specificity analysis (lines 413-415). In addition, the commercially available EUROIMMUN ELISA was validated by the manufacturer.

In 2020, the WHO, furthermore, established an international standard (human, NIBSC code: 19/178, WHO IS) consisting of sera from two MERS-convalescent humans to assess anti-MERS-CoV antibody responses and facilitate the comparability of MERS-CoV serological assays.⁴ In the study, a total of 22 serological assays conducted in 11 laboratories were assessed and compared using the IS, and, among them, the most frequently used assay was the commercial EUROIMMUN ELISA used in our study. In 2021, we analyzed the WHO IS in the EUROIMMUN ELISA and confirmed the reliable and precise detection of MERS-CoV-specific antibodies.

In our MVA-MERS-S trial, we assessed the comparability of the antibody assays and showed that

ID	SARS-CoV-2 S1 ELISA (IgG)		MERS-CoV ELISA (IgG) EUROIMMUN		
	WHO IS S1 BAU / ml	Result	OD value	Ratio*	Result
029-SARS-2	4727	positive	0.101	0.19	negative
030-SARS-2	3386	positive	0.033	0.06	negative
031-SARS-2	11513	positive	0.035	0.07	negative
032-SARS-2	611	positive	0.032	0.06	negative
033-SARS-2	4937	positive	0.039	0.07	negative
034-SARS-2	4169	positive	0.03	0.06	negative
035-SARS-2	6058	positive	0.016	0.03	negative
036-SARS-2	8117	positive	0.244	0.46	negative
037-SARS-2	8768	positive	0.029	0.05	negative
038-SARS-2	15870	positive	0.026	0.05	negative
039-SARS-2	633	positive	0.036	0.07	negative
040-SARS-2	500	positive	0.031	0.06	negative
041-SARS-2	721	positive	0.036	0.07	negative

IS=International Standard; BAU=binding antibody units. *A ratio of >1.1 is considered positive.

the OD values of both ELISAs as well as the in-house ELISA OD values and the reciprocal titer of neutralizing antibodies in the PRNT₈₀ correlated strongly (Koch et al., LID 2020, Figures 4D and Supplementary Figure S3, respectively)², as did anti-MERS-CoV reciprocal titers in immunofluorescence assay and PRNT₈₀ (Supplementary Figure 7 of this manuscript). We have now also included the correlation analysis of the MERS-CoV virus neutralization test (VNT) with the in-house ELISA in the Supplementary Appendix (Supplementary Figure 3) and likewise show a strong correlation (Spearman $r=0.78$, 95% CI 0.56-0.89, $p<0.0001$) despite a lower sensitivity.

Concerning the inclusion of individuals vaccinated against SARS-CoV-2, we would like to point out that such individuals were not included in the assays since this study was conducted until March 2019, well before the emergence of SARS-CoV-2. Cross-reactivity of the sera due to SARS-CoV-2 infection or vaccination can therefore be excluded. We, however, performed these analyses in the context of the currently ongoing Phase Ib trial of MVA-MERS-S-DF1 (NCT04119440). Below, we include the respective validation of the EUROIMMUN MERS-CoV ELISA. We analyzed sera from 13 individuals who had been vaccinated against COVID-19 with a SARS-CoV-2 S1 IgG ELISA and the MERS-CoV IgG EUROIMMUN ELISA for anti-SARS-CoV-2 and anti-MERS-CoV S1 antibody reactivity, respectively. While all individuals had positive anti-SARS-CoV-2 antibody titers, none showed reactivity in the anti-MERS-CoV ELISA (see Table 1).

Comment: In addition, the inclusion of control sera in the peptide microarray experiment would be useful to assess cross-reactivity of antibody responses to individual epitopes, which has particularly been observed for S2 epitopes. If possible, obtaining positive control sera from either human or dromedary camel cases of MERS-CoV would be useful to compare magnitudes of antibody responses to natural infection.

Reply:

We agree that the assessment of cross-reactivity of antibody responses to MERS-CoV (S2) epitopes and other human coronaviruses is highly interesting and of significance. Especially in the context of ongoing efforts to design universal betacoronavirus vaccine candidates, it will be particularly relevant to understand which epitopes may be possible candidates to elicit a cross-protective immune response.

To assure the specificity of antibody responses to the MERS-CoV epitopes we report here, we have included two kinds of control sera in the peptide microarray, as we have for the antibody assays discussed above. Sera of all vaccinees obtained pre-vaccination on Day 0 were included as intra-individual, paired controls, and antibody binding to the respective epitopes described significantly increased post-vaccination on B:D28. We, furthermore, assessed sera of an unvaccinated control participant with matching blood donation timepoints (D0, D28, D42, B:D0 and B:D28). We believe that the intra-individual controls are superior to the control of the unvaccinated individual, since they show the dynamic of antibody binding pre- and post-vaccination in the same sample group, and, therefore, included these in the main manuscript. However, to provide more information, we now also show the microarray including the unvaccinated individual's sera in the Supplementary Appendix (Supplementary Figure 4).

While we were unable to obtain sera from MERS-CoV convalescent dromedary camels or humans for this study due to lack of access, relevant data addressing antibody responses to MERS-CoV-S epitopes in natural infection have previously been published. Wang et al. performed a MERS-CoV spike protein peptide microarray using sera of convalescent human and dromedary camels as well as of uninfected human and camel controls.⁵ We have now included this point in the discussion (lines 307-312). The authors found that the immunogenic core region of the epitope that was recognized most frequently was AA 1230-1243, FQDELDEFFKNVS, located within the epitope AA 1225-1247 which we describe in our study. Antibodies targeting this epitope had previously been induced in mice after MERS-CoV-S immunization (see lines 304-307).

References:

- 1 Crompton, P. D. *et al.* A prospective analysis of the Ab response to Plasmodium falciparum before and after a malaria season by protein microarray. *Proc Natl Acad Sci U S A* **107**, 6958-6963, doi:10.1073/pnas.1001323107 (2010).
- 2 Koch, T. *et al.* Safety and immunogenicity of a modified vaccinia virus Ankara vector vaccine candidate for Middle East respiratory syndrome: an open-label, phase 1 trial. *Lancet Infect Dis* **20**, 827-838, doi:10.1016/S1473-3099(20)30248-6 (2020).

- 3 Okba, N. M. A. *et al.* Sensitive and Specific Detection of Low-Level Antibody Responses in Mild Middle East Respiratory Syndrome Coronavirus Infections. *Emerg Infect Dis* **25**, 1868-1877, doi:10.3201/eid2510.190051 (2019).
- 4 Mattiuzzo, G. *et al.* Establishment of 1st WHO International Standard for anti-MERS-CoV antibody. *WHO EXPERT COMMITTEE ON BIOLOGICAL STANDARDIZATION* (2020).
- 5 Wang, C. *et al.* A conserved immunogenic and vulnerable site on the coronavirus spike protein delineated by cross-reactive monoclonal antibodies. *Nat Commun* **12**, 1715, doi:10.1038/s41467-021-21968-w (2021).

Reviewers' Comments:

Reviewer #1:

Remarks to the Author:

The revised manuscript submitted by Fathi et al addresses all critiques raised in the initial review

Reviewer #2:

Remarks to the Author:

I thank the authors for their detailed responses to the initial review, particularly the inclusion of references to and discussion of additional negative and positive controls. While the separation of the long-term follow-up data into another manuscript is suboptimal, this should not prevent publication of the compelling data presented in this manuscript supporting the utility of a booster dose for this vaccine. I believe the edited manuscript is now suitable for publication.

Point-by-point response to reviewers' comments
Manuscript No.: NCOMMS-22-05187A

Reviewers' comments

Reviewer #1:

Comment: The revised manuscript submitted by Fathi et al addresses all critiques raised in the initial review

Reply: We are very pleased to read that the reviewer considers that the critiques raised have been adequately addressed. We would like to again thank the reviewer for their helpful comments and suggestions to include further safety data.

Reviewer #2:

Comment: I thank the authors for their detailed responses to the initial review, particularly the inclusion of references to and discussion of additional negative and positive controls. While the separation of the long-term follow-up data into another manuscript is suboptimal, this should not prevent publication of the compelling data presented in this manuscript supporting the utility of a booster dose for this vaccine. I believe the edited manuscript is now suitable for publication.

Reply: We are very happy to read that the reviewer believes the revised manuscript is now suitable for publication. We recognize that the addition of long-term follow-up data would have added another interesting aspect to the work presented here. We are thankful that the reviewer is nevertheless understanding and does not see the separation from the long-term follow-up study as an obstacle to publication of the data presented here. We would also like to thank the reviewer for their thought-provoking input during the review process.